# 3D printing direct to industrial roll-to-roll casting for fast prototyping of scalable microfluidic systems

**Amber L. Boutiette**[1], **Cristoffer Toothaker**[1], **Bailey Corless**[1], **Chouaib Boukaftane**[2], **Caitlin Howell**[1,3]*

**1** Department of Chemical and Biomedical Engineering, University of Maine, Orono, Maine, United States of America, **2** Sappi North America, Inc., Westbrook, Maine, United States of America, **3** Graduate School of Biomedical Science and Engineering, University of Maine, Orono, Maine, United States of America

* Caitlin.howell@maine.edu

**Data Availability Statement:** All quantitative data files are available as Supporting information.

**Funding:** This work was financially supported by and conducted in collaboration with Sappi North

## Abstract

Microfluidic technologies have enormous potential to offer breakthrough solutions across a wide range of applications. However, the rate of scale-up and commercialization of these technologies has lagged significantly behind promising breakthrough developments in the lab, due at least in part to the problems presented by transitioning from benchtop fabrication methods to mass-manufacturing. In this work, we develop and validate a method to create functional microfluidic prototype devices using 3D printed masters in an industrial-scale roll-to-roll continuous casting process. There were no significant difference in mixing performance between the roll-to-roll cast devices and the PDMS controls in fluidic mixing tests. Furthermore, the casting process provided information on the suitability of the prototype microfluidic patterns for scale-up. This work represents an important step in the realization of high-volume prototyping and manufacturing of microfluidic patterns for use across a broad range of applications.

## Introduction

Some of the most high-impact future markets for the development of microfluidic technology are those that will require high-volume production, such as microfluidic point-of-care (POC) diagnostics which could facilitate decentralized testing of patients and faster time-to-result in coordinated responses to outbreaks of disease [1] or the move toward personalized medicine [2–4]. Recent developments in droplet microfluidics, cell sorting, and organ-on-a-chip concepts have increased excitement over the potential of microfluidic systems [5]. However, there is a striking and widely recognized disconnect between the number of microfluidic applications being developed in academic labs and the number of microfluidic technologies being translated to the market [3, 6–10].

The major factors holding back microfluidic technology commercialization are a lack of standardization of components and the difficulty of scaling-up the fabrication approaches most widely used in academic research labs [5, 7, 11, 12]. The use of fabrication techniques such as soft lithography [13–15], or etching [16] followed by casting into polydimethylsiloxane (PDMS) [17], is broadly used in academia [10], but often results in relatively few replicates of a custom microfluidic design which cannot provide the statistical rigor necessary to justify commercial

America, Inc. The funder provided support in the form of salaries for authors CB and student support (stipend + tuition) for ALB, as well as funds for materials and supplies. The funder provided access to the industrial roll-to-roll casting equipment and employees, including CB, to run the equipment and troubleshoot. The funder did not play any role in the study design, analysis, interpretation of the data, writing of the paper, or the decision to submit for publication.

**Competing interests:** This work was financially supported by and conducted in collaboration with Sappi North America, Inc. The funder provided access to the roll-to-roll casting equipment and employees, including CB, to run the equipment and troubleshoot, as well as funds for materials and supplies. The funder did not play any role in the study design, analysis, interpretation of the data, writing of the paper, or the decision to submit for publication. This affiliation and collaboration does not alter our adherence to PLOS ONE policies on sharing data and materials.

investment [7], and in addition cannot be easily translated to mass-manufacture in original PDMS material [5]. Microfluidic devices fabricated from materials such as poly(methyl methacrylate) (PMMA), which could be mass-manufactured using injection molding [10] as a standardized design, often require the use of expensive dies and masters as well as the fabrication equipment itself [18], which can be beyond the financial capacity of many research groups.

Both academically-favoured PDMS moulding and industrially-favoured embossing/printing methods of manufacturing microfluidics do not generally methods which permit fast prototyping of prospective designs which can be immediately transferred to at-scale manufacture. Additive manufacturing or 3D printing of microfluidic has recently emerged as a potential solution which makes the fabrication process easier and more adaptive [13, 14, 19–21]. However, mass-manufacture using this technique is challenging due to limitations on available resolution of printers, throughput rate, and compatibility of resins [22]. Roll-to-roll fabrication strategies can offer near-immediate high-throughput and have been used to create functional microfluidic devices [23–25]; however, the creation of masters for use with this approach can be time-consuming and expensive, often requiring the use of cleanrooms [26, 27]. Furthermore, if a master created in this way does not work well due to unforeseen problems such as incomplete coating coverage, bubble formation, or poor release of the cast material, valuable time and money will have been lost. To facilitate the development of microfluidic devices which have a clear, direct route to industrial mass-manufacture, a method which combines both the efficient prototyping benefits of rapid prototyping with the low cost/high volume and automated production and standardization of roll-to-roll fabrication is required [5]. Importantly, early direct partnership with industry is critical when developing promising approaches in an academic setting to ensure eventual commercial feasibility [5, 7].

Here, we demonstrate a method which combines rapid prototyping with industrial-scale roll-to-roll manufacture to produce prototype microfluidic channels, permitting rapid, low-cost screening of channel patterns for both functionality and mass-manufacturability. For proof of concept, two-stream passive fluid mixers were chosen as the ability to rapidly mix two or more reagent streams is often a required function in microfluidic setups [28–32]. Four different designs with standardized inlets and outlets were 3D printed from digital files, then either directly molded in PDMS or coated with a thin layer of metal and shipped to Sappi North America, Inc., for roll-to-roll casting in an acrylated coating on their industrial equipment (Fig 1A). PDMS-cast and roll-to-roll cast patterns were then sealed using adhesive laminate sheets and placed into a reusable housing to be assessed for mixing efficiency with fluids (Fig 1B). Using the 3D print-to roll-to-roll method, multiple prototypes could be cast simultaneously by aligning the 3D printed masters side-by-side on the roll. Masters could be used for two successive passes before pattern degradation became significant, and provided valuable information about the feasibility of a high-throughput roll-to-roll process with the pattern of choice before a final pattern is selected for mass production. Quantitative analysis of the mixing index (a measure of mixing efficiency) revealed no differences between the PDMS-molded and roll-to-roll cast patterns, demonstrating the functionality of the 3D print to roll-to-roll prototypes. This approach provides a new tool to help bridge the gap between industry and academia in the drive to move microfluidic technology to market.

## Materials and methods

### Design generation and 3D printing

Four passive mixing patterns with various shapes were selected to demonstrate a dynamic range of geometrical features. Patterns were first designed using SolidWorks CAD modelling software (SolidWorks 2018–2019, Dassault Systems SOLIDWORKS Corp.) as inverse masters,

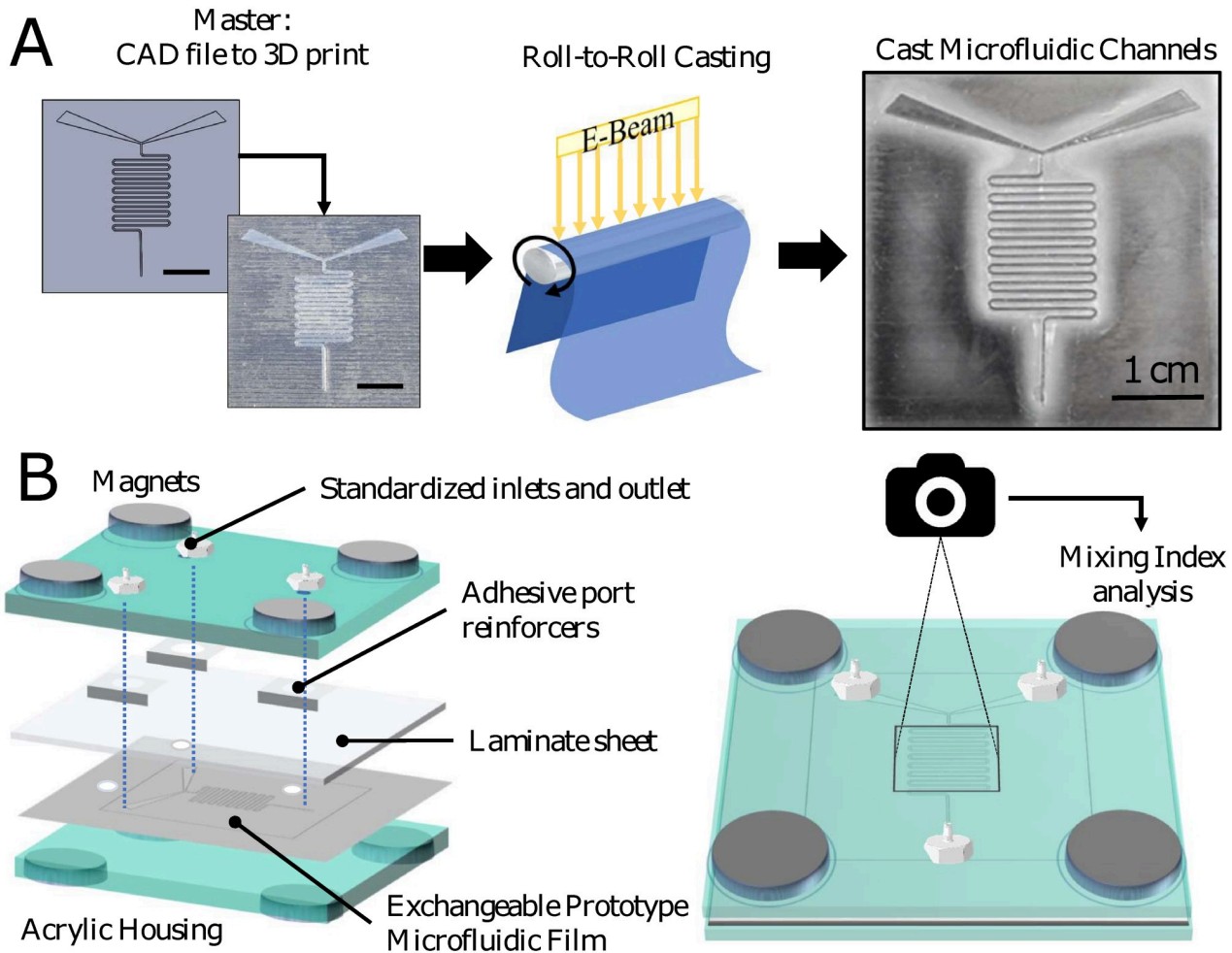

**Fig 1. Schematic of fabrication and experimental methods.** (A) A microfluidic pattern is first designed in CAD software then 3D printed (left, scale bar 1 cm). The pattern is then incorporated into an industrial-scale roll-to-roll continuous liquid casting process (center) to produce a cast microfluidic film (right). (B) Exploded view of the reusable microfluidic housing used to test the films (left) and top-view of mixing device with schematic representation of the analysis process (right).

then printed with an Objet30 Desktop 3D Printer with a resolution of ±100 μm. DurusWhite, a proprietary polypropylene-like material, was chosen as the printing material due to the flexibility it imparted to final printed parts, which prevented damage during pattern transfer in both the roll-to-roll casting and PDMS molding steps.

### Creation of PDMS and roll-to-roll microfluidic patterns

To create the PDMS microfluidic patterns, the 3D printed masters were adhered directly to the bottom of a petri dish using epoxy (Elantas Easypoxy® K-230). The prepolymer and curing agent (Sylgard 184 Silicone Elastomer Curing Agent and Base) were mixed in a ratio of 10:1, respectively, then mixed at 2000 rpm for 1 minute with a planetary centrifugal mixer (Thinky USA). The polymer mix was poured over the master and placed in a vacuum chamber for 60 minutes to degas. The samples were then cured at 70˚C for 60 minutes. The dish was removed and the PDMS mould was separated from the 3D print.

To create the roll-to-roll cast microfluidic film patterns, the 3D printed masters were first sputter-coated with a 35 nm layer of gold-palladium, then delivered to Sappi North America,

Inc., in Westbrook, ME, for use as masters in their proprietary electron beam (E-beam)-assisted roll-to-roll liquid casting process. This process consists of coating an acrylated polymer layer onto a transparent film backing pressing the master into the polymer layer in a roll-to-roll system, then curing the polymer against the master using an E-beam. This process is currently used on an industrial scale to produce release patterns for use in the textile industry and at full capacity can produce 4,200 $m^2$ of patterned material per hour. For this work, the sputter coating of the 3D printed patterns with metal was an essential step which prevented immediate degradation of the master upon exposure to the E-beam.

## Surface and pattern analysis

Contact angles of the cast microfluidic film material were assessed via water contact angle. Briefly, a 30 µL droplet of deionized water [33, 34] was placed on top of a planar section of the cast film and images taken with an EOS Rebel T5 camera (Canon). A total of seven replicates were imaged. Contact angle was calculated in ImageJ using the low-bond axisymmetric drop shape analysis plug-in [35]. Profilometry measurements were taken on an Alicona Infinite Focus (Bruker Alicona).

## Mixer assembly

A standardized device housing was designed to allow rapid assembly and disassembly of a closed microfluidic channel system for both cast film and PDMS mixing patterns and comprised laser-cut acrylic sheets with permanently fastened barbed adapters for tubing at the inlets and outlet. All microfluidic patterns were cleaned with isopropanol and air dried prior to assembly. Adhesive sheets (Fellowes 3-mil Self-adhesive sheets) were pierced with a 5.0 mm biopsy punch (World Precision Instruments) at the location of the inlets and outlets and placed over the microfluidic pattern substrate. Adhesive tabs (Scotch 12.7 mm×12.7 mm Mini Tabs) were also pierced with the 5.0 mm biopsy punch and placed with holes aligning those on the adhesive sheet. This unit was aligned with the inlets of the cast films, and all layers held in place and provided additional even pressure distribution by 4 sets of magnets (McMaster-Carr Twist-release paired magnets) arranged around the 4 corners of the device (Fig 1B). Tubing was attached between the inlet adaptors and two syringe pumps (New Era Pump Systems, Inc.), as well as to the outlet adaptor and a beaker to exhaust fluid. For each experiment, a fresh pattern with attached laminate sheet and adhesive tabs was swapped out in the acrylic housing. The entire unit was housed in a 30"×30" light tent (Westcott Digitent) for imaging to reduce the reflection of light and evenly illuminate the microfluidic setup.

## Fluid mixing tests

Quantitative analysis of fluid mixing was conducted for the serpentine patterns by flowing two solutions into the microfluidic channels with the aid of syringe pumps. A 3 ml syringe containing deionized water and a second 3 ml syringe containing a 1% mixture of black ink (Higgins) and water were mounted on syringe pumps (New Era Pump Systems, Inc.) and dispensed at 0.15 ml/min. This flow rate was chosen as it resulted in mixing occurring over a time scale that was quantifiable using our camera setup. The system was run for 60 seconds to ensure equilibrium was reached before images for each trial were taken. Digital images of the flow were obtained using the EOS Rebel T5 digital camera with a 0.25 m Macro Lens mounted on a post and stand. At least 5 independent samples were analysed for both the film and PDMS patterns.

## Image analysis

Mixing indices were calculated from the images gathered in the mixing analysis by assessing the intensity values of pixels across a section of a gray-scale image where a mixing event had occurred. All indices involved metrics of the standard deviation of pixel intensities to quantify the mixing profile, as a more homogeneous and thoroughly mixed sample will generally have a low standard deviation in pixel intensity while a less thoroughly mixed sample will have a higher standard deviation. The absolute mixing index (AMI) is a method of comparing the standard deviation of pixel intensities to the mean intensity value for a more direct measure of the extent of mixing [36] and is calculated using the formula:

$$AMI = \frac{\sigma}{\langle I \rangle} = \frac{\sqrt{\frac{1}{N} \Sigma_{i=1}^{N} (I_i - \langle I \rangle)^2}}{\langle I \rangle} \tag{1}$$

where $I_i$ is the local pixel intensity, $\langle I \rangle$ the average of the pixel intensities in the cross section, $N$ the total number of pixels. The experimental images were analysed using ImageJ. Images were converted to 8-bit and pixel intensity measurements were made at the cross sections of all locations of interest on the serpentine channels using the line tool. Intensity values for the pixels along this line were extracted and each set of pixel values were normalized to the same span of intensities as 0–255 on a gray-scale image. These values were then exported into Microsoft Excel, where the AMI formula was applied to each set to calculate the absolute mixing intensity for each location of interest along the length of the channel. The absolute mixing indices with modified intensities were plotted.

## Statistical analysis

Data were used as generated with no pre-processing (e.g. transformation, normalization, evaluation/removal of outliers). Quantitative data are presented as meant ± standard deviation. At least 5 independent replicates were evaluated for the fluidic mixing tests, and 7 for the contact angle analysis.

## Results and discussion

Fig 2A shows an exploded schematic of a roll-to-roll cast microfluidic film created from a 3D printed master. The acrylated polymer overlayer coats the entire surface of the backing, creating a water-tight layer of polymer with a contact angle of 95.1˚± 3.1˚ (n = 7, Fig 2B). Using the 3D print-to roll-to-roll casting process outlined in Fig 1, several different microfluidic patterns were fabricated: a serpentine pattern (Fig 1A) developed by Romanov et al., [19] a planar spiral from work by Duryodhan et al (Fig 2C, left) [32], a semicircle split/recombine pattern as described by Sivashankar et al. (Fig 2C, center) [37], and an original off-center alternating diamond pattern (Fig 2C, right). For initial proof-of-concept tests, each pattern consisted of standardized wide entry ports spaced 3.5 cm apart, narrowing down into channels ~250 μm deep with sloped walls measuring ~600 μm at the widest point. The process of roll-to-roll casting with the relatively flexible 3D printed master resulted in a thicker layer of white acrylated coating material directly adjacent to the channels and a thinner layer farther from the channels, presumably where the master was pressed closer to the film surface. For fast prototyping, several 3D printed masters could be lined up side-by-side to produce multiple microfluidic patterns simultaneously on the 71 cm-wide role (Fig 2D), with a new pass occurring every 64 cm. The final cast film patterns consisted of very little total material, rendering them quite flexible (Fig 2E). The final dimensions of the cast films were found to be very close to the value for the 3D printed master, which measured 500 ± 100 μm across and 390 ± 100 μm deep, but less than

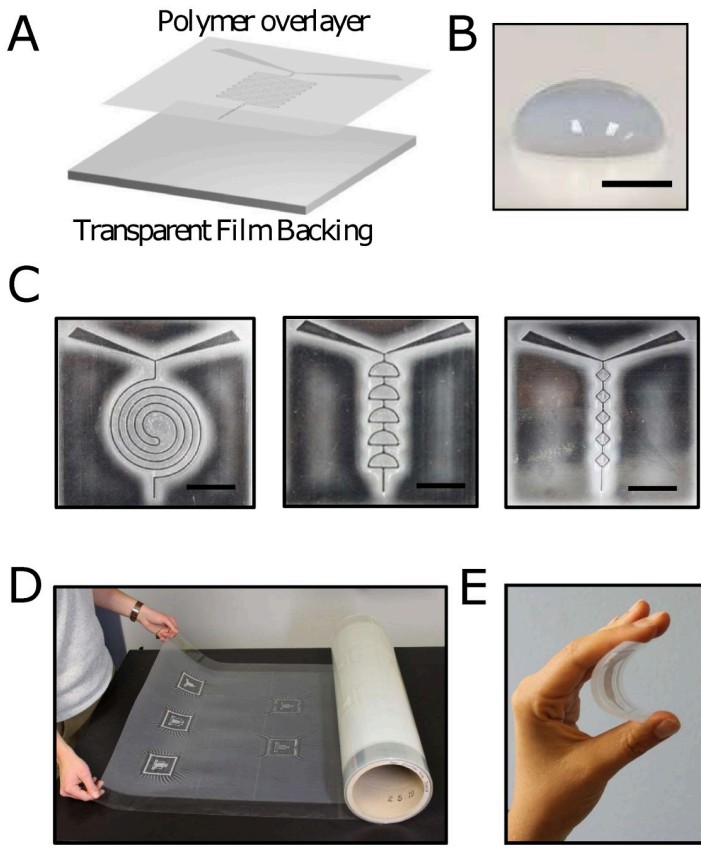

**Fig 2. Roll-to-roll cast channel systems.** (A) An exploded schematic view of an acrylated polymer overlayer on a transparent film backing (B) A water droplet on an unpatterned area of the film demonstrating the hydrophobic nature of the coating. Scale bar = 2 mm. (C) Three types of microfluidic mixer patterns cast on film using the roll-to-roll process. Scale bar = 1 cm. (D) A cast roll showing one complete pass resulting in five mixer prototypes from five 3D printed masters of the types shown in part (C). (E) A demonstration of the flexibility of the patterns when cast on film.

the original CAD design, which defined both the channel width and depth as 400 μm. The difference in widths between the CAD design and the final cast films are therefore more likely to be due to the variability in 3D printed master, as these features were near the lower limit of resolution of the printer. Continuous advances in 3D printing technology have already resulted in prints with features down to the single-μm scale [38], and roll-to-roll lithography is known to be capable of replicating sub-micron patterns with high fidelity [39]. The results presented here therefore suggest that patterns with smaller features could also be successfully replicated, as long as a suitable master could be produced.

To establish the functionality of the roll-to-roll cast microfluidic devices, we compared them to devices fabricated in the traditional manner: PDMS casting. For these tests, serpentine microfluidic channel systems were chose, as they are widely-used and well-known in the field of microfluidics [40, 41]. Both the roll-to roll devices and the PDMS devices were cast from the same 3D printed masters, and both channel systems were sealed with laminate sheets and loaded into the reusable microfluidic housing shown in Fig 1B. As others have noted [42], care had to be taken in sealing the channel systems with the laminate sheets to avoid leaks. Water-soluble black ink and pure water were then introduced into the channels at 0.15 mL/min and the system allowed to reach equilibrium (Fig 3A). Ink and clear water were chosen to provide

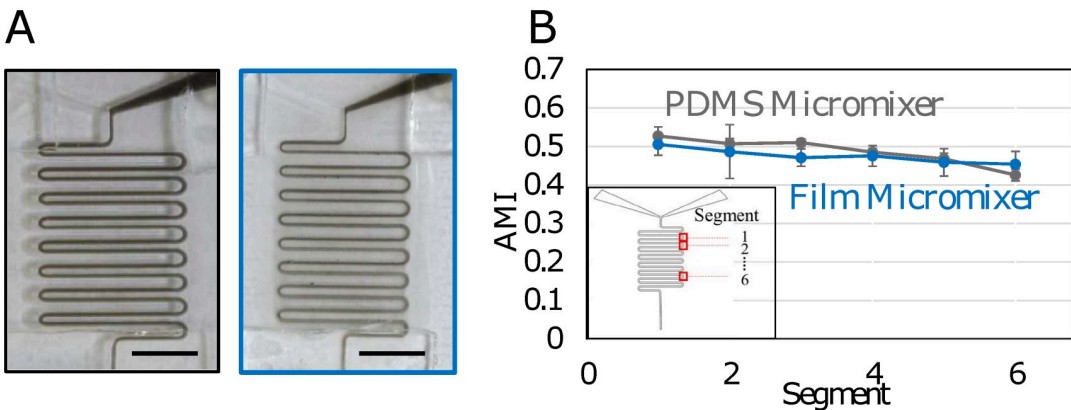

**Fig 3. Fluidic mixing of roll-to-roll and PDMS cast devices.** (A) Sample images of both PDMS (left, black border) and film (right, blue border) mixing experiments, showing successful fluid flow in microchannels. (B) Plot of absolute mixing index (AMI) vs segment location along length of microchannel for the PDMS (dark grey) and film (blue) micromixer (n = 5). Inset: Geometry of serpentine mixer with boxes indicating locations of image analysis and mixing quantification.

the largest contrast possible for quantitative image analysis. To quantify the performance of the micromixers, the absolute mixing index (AMI) was calculated at 6 turns along the serpentine mixer. The first and last turns were not analysed due to interference from the adhesive entry/exit port reinforcers. Five trials were run of both cast film and PDMS micromixer patterns and the average AMI values were plotted as a function of increasing distance from the liquid inlets (Fig 3B). A value of 0 on the plot indicated complete mixing, while a value of 1 indicated no mixing [36]. For both cast film and PDMS patterns, the mixing index values of both the PDMS and cast patterns decreased from slightly above 0.5 to around 0.45 along the length of the channel, indicating a slight increase in mixing in both channel types without reaching a fully mixed state.

In microfluidic channels with straight planar geometries, mixing occurs purely by diffusion [32]. In curved channels like those used in the mixing experiment, transverse secondary Dean flows arise due to the interaction between centrifugal and inertial forces. The curved geometries used in the serpentine mixer enhance these secondary flows, causing the fluid to travel from the outer to the inner regions where the radius of curvature is smallest [43]. Since the direction of rotation of the secondary flows is not sustained over the length of the mixer it is expected that their strength would not be significant enough to perturb the laminar profile, and over a shorter mixing length the primary mixing mechanism would still be diffusion. For this reason, it was expected that fluids would not reach complete mixing within the channel length of the serpentine mixers. This was confirmed by the mixing index results, which showed ranges that remained above 0.30 for both the film and PDMS patterns. Nevertheless, as these values also show significant areas of overlap, it is reasonable to conclude that mixing performance in the film-based channels is comparable to that in the industry-standard PDMS channels. Discrepancies in values are likely due to the different thickness and optical properties of the PDMS and film patterns: the roll-to-roll cast films had an increase in opacity in the regions immediately surrounding the mixing channel, while the PDMS channels were evenly transparent. This may have resulted in variations in the adjusted focus length of the camera between mixing trials, ultimately resulting in slight variations in pixel resolution.

To test the durability of the 3D printed masters in roll-to-roll casting, a new set of masters were run for multiple consecutive passes through the coating and E-beam curing process. Fig 4A shows the films cast in the 1st, 2nd, 3rd, and 68th pass. This number of passes was chosen

as it allowed for a ramp up, run, and ramp down of the machinery. The new masters were designed with a raised border around the edge of the microfluidic channels to more tightly define the height surrounding the channels, in contrast to the previous masters without borders (Fig 2C). The 1st pass showed a well-formed channel, with the appearance of a few bubbles in between the channels. In a prototyping process with the intent to identify an optimal design for mass-manufacture, the presence of bubbles such as these would indicate that the change that had been made (i.e. moving from no border to a fitting border) may present problems in a final design, and so a different border should be considered despite the fact that the channel system itself was still intact. Such information may not have been readily available if only a bench-scale fabrication process had been used, and could have resulted in an investment of time and resources into optimizing a pattern that would likely require a redesign for scale-up.

The 1st pass of the 3D printed master showed what appeared to be residuals of the metal coating on the cast film itself. Films cast in the 2nd pass showed the appearance of thinning areas of the acrylated coating making up the raised border but were still functional. By the 3rd pass, the border material is nearly completely gone, with the exception of those areas immediately adjacent to the channel walls. This appearance continued to be produced through multiple successive passes, up to and including the 68th pass, shown in the lower right of Fig 4A. Profilometry measurements of the first and 68th pass at the convergence of the two inlet channels, shown in Fig 4B, confirmed the loss of material from the surrounding border but retention of the material immediately next to the channel itself. However, the channels were not completely unchanged by the multiple passes of the master through the rolls: measurement of the channel showed a maximum channel depth of 269 µm for the 1st pass, but only 209 µm by the 68th pass.

The changes observed in the cast films from the 1st to the 2nd to the 3rd pass, which was observed in multiple similar runs, could be explained by an accumulation of the acrylated coating material on the 3D printed master itself. Ionizing radiation from the E-beam and slight deformation of the master caused by the rolls may have contributed to this process, which likely began with the removal of the protective metal coating in the 1st pass, followed by

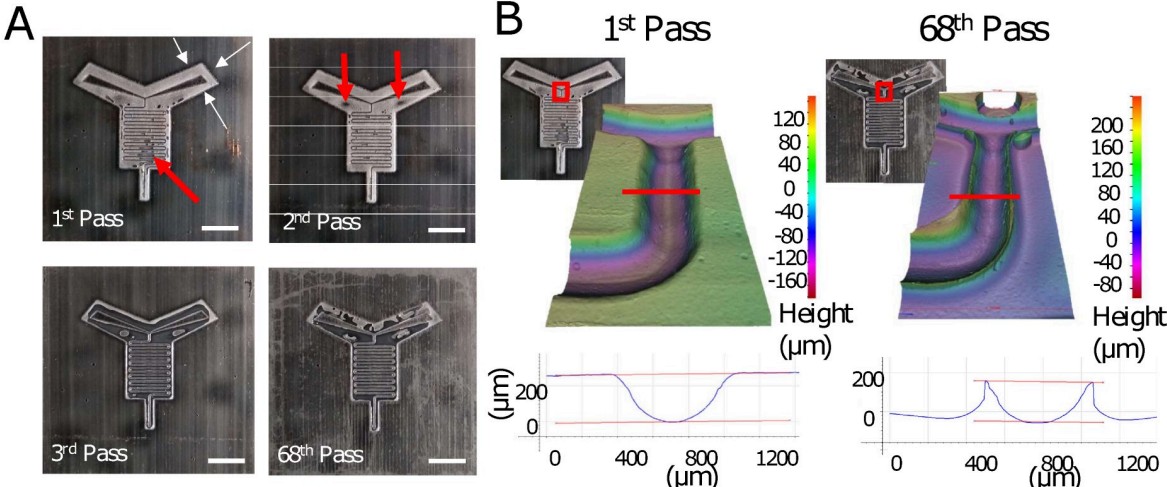

**Fig 4. Results of successive passes using a 3D printed master.** (A) Photos of the films cast from the 1st, 2nd, 3rd, and 68th pass of the 3D master over the roll. White arrows indicate the raised border around the channels. Red arrows in the picture of the 1st pass indicate bubble formation, while those in the 2nd pass indicate areas where the coatings is noticeably thinner. Scale bars = 1cm. (B) Profilometry measurements of the first and 68th pass. Picture insets with red boxes indicate the location of the 3D profilometry measurements, while red lines on the 3D plots indicate where the cross-sectional measurements (below) were taken.

localized accumulation in the 2nd pass, and finally full-surface coverage of the coating by the 3rd pass. The observation that the channels remain for multiple passes may be due to the fact that the non-horizontal surfaces of the 3D printed master were not exposed to the E-beam directly and were also less likely to be deformed. This may have limited the ability of the acrylated coating to adhere within the channels themselves. Yet despite the fact that channels were still present, use of the cast patterns was compromised by the lack of a planar border onto which the laminate sheets could be attached to enclose the channel system. This prevented testing the functionality of the films formed in the latter passes. However, the results suggest that increasing the adhesion strength and/or thickness of the metal coating layer as well as the selection of a polymer with higher radiation tolerance and durability, may increase the number of passes that the 3D printed masters can sustain if required. In the setup used to cast the films shown here, a total of 8 masters could have fit across the roll and another 5 around the circumference, which would have yielded a total of 80 functional patterned films from the first two passes with no master stability modifications, which in some applications may be sufficient for preliminary bench testing. As mounting new patterns masters on the roll for intended for a low number of passes (e.g. 50–60) can be done relatively simply, for example with aluminium tape, the labor associated with changing the masters on the roll is minimal. Once an optimal pattern has been identified that is both functional and compatible with the mass-production process, a metal master could then be created to permit larger-scale manufacture.

Suitability for mass-manufacture and standardization of components are two of the major challenges standing between the development of microfluidic solutions and their deployment as commercial products [7, 10, 12]. This work seeks to help address these challenges by demonstrating a method by which prospective patterns can be created via 3D printing masters—with a standard digital file shared among researchers and members of industry—then tested for mass-manufacturability early-on in an industrial roll-to-roll process. This approach could help to streamline scale-up by helping researchers to focus their development efforts on microfluidic patterns with a known performance in a mass-manufacturing process, and could help industry adoption by providing a greater certainty of the possibility of production at the volumes necessary to reach an economy of scale. Once a final design is optimized, the 3D printed masters can be exchanged for more durable metal masters for high-volume replication. The volume of production that is possible to achieve with the industrial roll-to-roll method used here (4,200 $m^2$ of cast material per hour) would likely be sufficient to produce disposable microfluidics, a step that has been identified as necessary in many medical applications to avoid cross-contamination [12]. Nevertheless, it should be noted that this approach may be limited to particular polymers in terms of the material that can be successfully cast. This may require changes to traditional experimental setups such as using different fluidic connectors and alternative methods of sealing the channel systems. However, as long as these differences are taken into account, the 3D printing direct to roll-to-roll casting method described here may help to reduce costs and streamline the prototyping process of industrially-produced microfluidic devices.

## Conclusions

We have demonstrated a new method of rapidly prototyping microfluidic channel designs that simultaneously permits analysis of functionality in PDMS and allows assessment of mass-manufacturability in an industrial-scale roll-to-roll E-beam process. Microfluidic channel pattern masters were 3D printed, then coated with a thin layer of metal to improve durability in the industrial-scale casting process. The metal-coated 3D masters were then used to produce roll-to-roll cast films with channel system embedded in an acrylated polymer coating. Using

this method, a variety of different microfluidic film prototypes could be quickly produced, then roll-to-roll cast simultaneously and analysed for both functionality and suitability for scale-up. Analysis of the absolute mixing index of the roll-to-roll cast film devices and the PDMS devices run with two fluids showed no significant differences in mixing efficiency between the cast and PDMS channel systems. Finally, the 3D printed masters were subjected to multiple passes through the roll-to-roll process to determine durability. It was found that with the materials used, patterns could be run for two complete passes before significant pattern degradation occurred. These tests also demonstrated the usefulness of the fast prototyping process, as the master pattern selected resulted in the appearance of bubbles in the roll-to-roll casting, providing information on the suitability of the pattern for scale-up which would not have been possible to obtain using traditional bench-level fabrication processes. The ability to use 3D printed masters in industrial-scale roll-to-roll manufacturing processes may prove useful in bridging the gap between microfluidic technology development and translation to the market.

## Supporting information

**S1 File. Profilometry data for the 1$^{st}$ pass.**
(PDF)

**S2 File. Profilometry data for the 2$^{nd}$ pass.**
(PDF)

**S3 File. Spreadsheet data for the absolute mixing index calculations.**
(PDF)

## Acknowledgments

The authors thank Amy Blakeley and Nicholas Yardy for helpful discussions and invaluable advice, as well as Allen Treadwell of the UMaine Advanced Manufacturing Center, Kelly Edwards of the Coordinated Operating Research Entitites, and Mark Philbrick of Sappi North America, Inc., for assistance with preparation and running of the masters. The authors also thank Adrian Arias Palomo for technical editing.

## Author Contributions

**Conceptualization:** Amber L. Boutiette, Caitlin Howell.

**Data curation:** Amber L. Boutiette, Caitlin Howell.

**Formal analysis:** Amber L. Boutiette, Caitlin Howell.

**Funding acquisition:** Caitlin Howell.

**Investigation:** Chouaib Boukaftane, Caitlin Howell.

**Methodology:** Cristoffer Toothaker, Bailey Corless, Caitlin Howell.

**Project administration:** Caitlin Howell.

**Resources:** Caitlin Howell.

**Supervision:** Caitlin Howell.

**Validation:** Caitlin Howell.

**Visualization:** Caitlin Howell.

**Writing – original draft:** Amber L. Boutiette, Caitlin Howell.

**Writing – review & editing:** Caitlin Howell.

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
