## [Decision Letter · Decision Letter 0]

21 Sep 2020

PONE-D-20-20639

3D Printing Direct to Industrial Roll-to-Roll Casting for Fast Prototyping of Scalable Microfluidic Systems

PLOS ONE

Dear Dr. Howell,

Thank you for submitting your manuscript to PLOS ONE. The manuscript has been seen by two referees. Both of them have identified multiple issues with experimental design and methodology. After careful consideration, we feel that it has merit but does not fully meet PLOS ONE’s publication criteria as it currently stands. Therefore, we invite you to submit a revised version of the manuscript that addresses the points raised during the review process. If you think that addressing all the points requires considerable time, we will be happy to consider a new submission from you at some point in future.

We look forward to receiving your revised manuscript.

Kind regards,

Ivan Kryven

Academic Editor

PLOS ONE

Journal Requirements:

"C.H. received funding from Sappi North America, Inc. to conduct this work (https://www.sappi.com/region/north-america). The funder had no role in the study design and preparation of the manuscript. Profilometry measurements were conducted at Sappi Westbrook and the funder reviewed the manuscript and approved it prior to submission."

We note that one or more of the authors have an affiliation to the commercial funders of this research study : Sappi North America, Inc..

2.1. Please provide an amended Funding Statement declaring this commercial affiliation, as well as a statement regarding the Role of Funders in your study. If the funding organization did not play a role in the study design, data collection and analysis, decision to publish, or preparation of the manuscript and only provided financial support in the form of authors' salaries and/or research materials, please review your statements relating to the author contributions, and ensure you have specifically and accurately indicated the role(s) that these authors had in your study. You can update author roles in the Author Contributions section of the online submission form.

2.2. Please also provide an updated Competing Interests Statement declaring this commercial affiliation along with any other relevant declarations relating to employment, consultancy, patents, products in development, or marketed products, etc.  

Reviewers' comments:

Reviewer's Responses to Questions

**Comments to the Author**

1. Is the manuscript technically sound, and do the data support the conclusions?

Reviewer #1: Yes

Reviewer #2: Yes

2. Has the statistical analysis been performed appropriately and rigorously? 

Reviewer #1: Yes

Reviewer #2: Yes

3. Have the authors made all data underlying the findings in their manuscript fully available?

Reviewer #1: Yes

Reviewer #2: Yes

4. Is the manuscript presented in an intelligible fashion and written in standard English?

Reviewer #1: Yes

Reviewer #2: Yes

5. Review Comments to the Author

Reviewer #1: PONE-D-20-20639 “3D Printing Direct to Industrial Roll-to-Roll Casting for Fast Prototyping of Scalable Microfluidic Systems” compares the replication performance of an injet 3D printed device cast in PDMS and replicated by roll casting. The manuscript is well positioned and clearly identifies the capability gap of rapid prototyping suing a manufacturing approach that is compatible with mass manufacturing. The manuscript, however, falls short in demonstrating their solution and potential reasons for that are followed-up insufficiency.

It is impressive that 68 passes of roll casting were evaluated, but considering serious deterioration was observed in the second and third pass, I am unsure what the 68 runs add, rather than that the deterioration in the firs three was probably an unexpected outcome. In a manuscript like this, and considering eh rapid prototyping in design and materials of 3D printing, I would have expected a comparison of different materials, and the introduction of one edge in an (non-conclusive) attempt to fix the repeatability issue falls short delivering on the development of design features.

So the comparable mixing performance is a great start, and I truly believe the approach or using 3D printed templates for rollcasting is valid. But, before publishing, I suggest selecting a range of materials of different hardness (and adhesion of the metal) in order to identify one where a batch of can be replicated 10 times. Otherwise, you are as well off using 3 D printing only.

In order to assess if 10 (or 15, perhaps 8?) replication cycles are required, the authors should include a discussion on the labour and manual handling involved in replacing one template with another.

Reviewer #2: This manuscript titled in “3D Printing Direct to Industrial Roll-to-Roll Casting for Fast Prototyping of Scalable Microfluidic Systems” develops and validates a method to create functional microfluidic prototype devices using 3D printed masters in an industrial-scale roll-to-roll continuous casting process. I believe the work will be of interest to many in bridging the gap between microfluidic technology development and translation to the market. Before considering this manuscript for publication, the authors should consider the following points in any revision as follows:

1. The abstract is described abstractly without specific description of the work. It is suggested to modify the conclusion to emphasize the highlights and workload of the article.

2. In the introduction, the authors should introduce the advantage of the technology in this work in more detail.

3. How about the mechanical strength of PDMS?

4. In Fig.2C, three types of microfluidic mixer patterns are prepared. How about their absolute mixing index?

5. The authors should perform the structure of microfluidic channels by SEM.

6. Why both the channel width and depth are 400 μm? Has this been optimized?

7. This manuscript proposes a new method. There are many schematic diagrams and renderings in the manuscript, lacking effective data support and specific experiments to prove the possibility of relevant applications, such as the mixing effect of various fluids.

8. The size of 3D printing is very critical. It is suggested to indicate the scale and important information on the legend.

9. The authors should check the manuscript carefully to avoid the basic mistakes. For example, flexible typesetting of lines 18-21 in page 9, indent the first line by the same distance, space between numbers and units, etc. The above questions are focused on page 9-10.

10. In page 9 and 10, what is the purpose of gold-plated particles mentioned many times by the authors?

11. What is the effect of E-beam in the manuscript?

12. In page 13, the authors test the hydrophilic and hydrophobic surface of the film, but the volume of the droplets added by the authors is 30 microliters, and the authors should use smaller droplets.

13. In Fig. 2D and 2E, the authors should provide a scale-bar for the photo to determine the size of the device?

14. In the manuscript, the authors explore fluid mixing, and the flow rates of the two fluids should be explained.

I will be happy to recommend for publication a revised version of the manuscript in PLOS ONE.

6. PLOS authors have the option to publish the peer review history of their article (what does this mean?). If published, this will include your full peer review and any attached files.

Reviewer #1: No

Reviewer #2: No

---

## [Author Response · Author response to Decision Letter 0]

11 Nov 2020

Comments from the Editorial Office

Statistics: For original research, please check that your manuscript includes a sub-section entitled "Statistical Analysis" at the end of the Experimental Section that fully describes the following information: 1. Pre-processing of data (e.g., transformation, normalization, evaluation of outliers), 2. Data presentation (e.g., mean ± SD), 3. Sample size (n) for each statistical analysis, 4. Statistical methods used to assess significant differences with sufficient details (e.g., name of the statistical test including one- or two-sided testing, testing level (i.e., alpha value, P value), if applicable post-hoc test or any alpha adjustment, validity of any assumptions made for the chosen test), 5. Software used for statistical analysis. Figure legends: Please make sure that all relevant figure legends contain the information on sample size (n), probability (P) value, the specific statistical test for each experiment, data presentation and the meaning of the significance symbol.

Authors: We have added a dedicated Statistic section with all the relevant details requested here and have added in sample sizes in all relevant figure captions as well as the text.

 Authors: We have gone through all the formatting requirements and changed the manuscript accordingly.

 Authors: We have added more discussion of the advantage to the introduction.

Competing Interests Statement

This work was financially supported by and conducted in collaboration with Sappi North America, Inc. The funder provided access to the roll-to-roll casting equipment and employees, including CB, to run the equipment and troubleshoot, as well as funds for materials and supplies. The funder did not play any role in the study design, analysis, interpretation of the data, writing of the paper, or the decision to submit for publication. This affiliation and collaboration does not alter our adherence to PLOS ONE policies on sharing data and materials.

 Updated Funding Statement

3. Please provide an amended Funding Statement declaring this commercial affiliation, as well as a statement regarding the Role of Funders in your study. 

This work was financially supported by and conducted in collaboration with Sappi North America, Inc. The funder provided support in the form of salaries for authors CB and student support (stipend + tuition) for ALB, as well as funds for materials and supplies. The funder provided access to the industrial roll-to-roll casting equipment and employees, including CB, to run the equipment and troubleshoot. The funder did not play any role in the study design, analysis, interpretation of the data, writing of the paper, or the decision to submit for publication. 

3. Upon re-submitting your revised manuscript, please upload your study’s minimal underlying data set as either Supporting Information files or to a stable, public repository and include the relevant URLs, DOIs, or accession numbers within your revised cover letter. 

 Authors: We have uploaded the minimal underlying data sets as Supporting Information files.

---

## [Decision Letter · Decision Letter 1]

8 Dec 2020

3D Printing Direct to Industrial Roll-to-Roll Casting for Fast Prototyping of Scalable Microfluidic Systems

PONE-D-20-20639R1

Dear Dr. Howell,

We’re pleased to inform you that your manuscript has been judged scientifically suitable for publication and will be formally accepted for publication once it meets all outstanding technical requirements.

Kind regards,

Ivan Kryven

Academic Editor

PLOS ONE

Additional Editor Comments (optional):

Reviewers' comments:

Reviewer's Responses to Questions

**Comments to the Author**

1. If the authors have adequately addressed your comments raised in a previous round of review and you feel that this manuscript is now acceptable for publication, you may indicate that here to bypass the “Comments to the Author” section, enter your conflict of interest statement in the “Confidential to Editor” section, and submit your "Accept" recommendation.

Reviewer #2: All comments have been addressed

2. Is the manuscript technically sound, and do the data support the conclusions?

Reviewer #2: Yes

3. Has the statistical analysis been performed appropriately and rigorously? 

Reviewer #2: Yes

4. Have the authors made all data underlying the findings in their manuscript fully available?

Reviewer #2: Yes

5. Is the manuscript presented in an intelligible fashion and written in standard English?

Reviewer #2: Yes

6. Review Comments to the Author

Reviewer #2: The authors have carefully addressed all the issues I raised previously. I recommend it for publication in PLOS ONE.

7. PLOS authors have the option to publish the peer review history of their article (what does this mean?). If published, this will include your full peer review and any attached files.

Reviewer #2: No

---

## [Editor Report · Acceptance letter]

15 Dec 2020

PONE-D-20-20639R1 

3D Printing Direct to Industrial Roll-to-Roll Casting for Fast Prototyping of Scalable Microfluidic Systems 

Dear Dr. Howell:

I'm pleased to inform you that your manuscript has been deemed suitable for publication in PLOS ONE. Congratulations! Your manuscript is now with our production department. 

Kind regards, 

on behalf of

Dr. Ivan Kryven 

Academic Editor

PLOS ONE